# Flubendazole Enhances the Inhibitory Effect of Paclitaxel via HIF1α/PI3K/AKT Signaling Pathways in Breast Cancer

**DOI:** 10.3390/ijms242015121

**Published:** 2023-10-12

**Authors:** Yuxin Zhou, Minru Liao, Zixiang Li, Jing Ye, Lifeng Wu, Yi Mou, Leilei Fu, Yongqi Zhen

**Affiliations:** 1Department of Biotherapy, Cancer Center and State Key Laboratory of Biotherapy, West China Hospital, Sichuan University, Chengdu 610041, China; charlotte_zyx@outlook.com (Y.Z.); 15680965456@163.com (M.L.); scuyj0222@163.com (J.Y.); 18482273923@163.com (L.W.); skayhy@163.com (Y.M.); 2Sichuan Engineering Research Center for Biomimetic Synthesis of Natural Drugs, School of Life Science and Engineering, Southwest Jiaotong University, Chengdu 610031, China; li69010@163.com

**Keywords:** flubendazole, paclitaxel, hypoxia-inducible factor-1α, PI3K, AKT, combination therapy, breast cancer

## Abstract

Paclitaxel, a natural anticancer drug, is widely recognized and extensively utilized in the treatment of breast cancer (BC). However, it may lead to certain side effects or drug resistance. Fortunately, combination therapy with another anti-tumor agent has been explored as an option to improve the efficacy of paclitaxel in the treatment of BC. Herein, we first evaluated the synergistic effects of paclitaxel and flubendazole through combination index (CI) calculations. Secondly, flubendazole was demonstrated to synergize paclitaxel-mediated BC cell killing in vitro and in vivo. Moreover, we discovered that flubendazole could reverse the drug resistance of paclitaxel-resistant BC cells. Mechanistically, flubendazole was demonstrated to enhance the inhibitory effect of paclitaxel via HIF1α/PI3K/AKT signaling pathways. Collectively, our findings demonstrate the effectiveness of flubendazole in combination with paclitaxel for treating BC, providing an insight into exploiting more novel combination therapies for BC in the future.

## 1. Introduction

Breast cancer (BC) is by far one of the most prevalent and lethal cancers in women [1]. Chemotherapy remains the gold standard for treating BC currently. Chemotherapy drugs damage cancer cells in various ways, with some drugs acting by preventing cell division, while others cause cell death by damaging cell DNA [2,3]. Paclitaxel, as a representative first-line chemotherapy drug for BC, is believed to induce allergy and neuropathy at high doses, which is the main side effect of paclitaxel in cancer treatment [4,5]. Recurrence of metastatic breast cancer is characterized by rapid proliferation of drug-resistant cancer cells and high mortality. Chemotherapy alone cannot bring significant improvement and often leads to drug resistance in patients [6]. In advanced metastatic BC, drug resistance to paclitaxel can lead to a severe adverse prognosis, serving as a leading cause of mortality resulting from treatment failure [6]. Therefore, the primary global challenge of paclitaxel as the main anticancer chemotherapy drug to put into application is to reduce side effects and improve drug efficiency. Combination therapy is one of the most effective strategies for breast cancer treatment at present. More and more evidence has shown that the combination of chemotherapy and small molecule drugs has considerable efficacy in the treatment of breast cancer [7]. Therefore, it is of great significance to explore new synergistic drug combinations to overcome the side effects and drug resistance of paclitaxel in breast cancer cells.

Flubendazole, a benzimidazole-class anthelmintic, has been repurposed as an anticancer drug, owing to its extensive anticancer effects [8]. Several studies have demonstrated the antiproliferative effects of flubendazole on a variety of cancers [9,10,11,12]. Our previous studies have indicated that flubendazole is a potential chemical for the treatment of BC through regulating autophagy [13,14,15]. Moreover, a study has reported that combination therapy with mebendazole and docetaxel can act synergistically for prostate cancer treatment [16]. Furthermore, the paclitaxel and flubendazole combination decreases cell viability more effectively in intestinal cancer cells [17]. These findings collectively suggest that flubendazole may function as an effective paclitaxel synergist in the treatment of BC.

In this study, we initially employed a combination index (CI) calculation to evaluate the synergistic effects of paclitaxel and flubendazole in BC cells. Subsequently, the results of both in vitro and in vivo experiments demonstrated that flubendazole significantly enhances the cytotoxicity of paclitaxel against BC cells. Mechanistically, we found that flubendazole enhances the inhibitory effect of paclitaxel to BC cells via HIF1α/PI3K/AKT signaling pathways. Importantly, flubendazole was shown to reverse the drug resistance of paclitaxel-resistant BC cells. Overall, these findings offer valuable insights and potential avenues for exploring novel combination therapies for BC in future research.

## 2. Results

### 2.1. Flubendazole Enhances the Inhibitory Effect of Paclitaxel in BC Cells

To investigate the inhibitory effect of flubendazole in combination with paclitaxel on BC, we employed cytotoxicity assays. The results indicated that that the optimum combination ratio of flubendazole and paclitaxel is 3:1 with a concentration of 1.5 μM and 0.5 μM, respectively, based on the analysis of the CI and IC_50_ values (Figure 1A,B and Appendix A). Accordingly, a concentration of 1.5 μM flubendazole, 0.5 μM paclitaxel, and a combined application group of 1.5 μM flubendazole and 0.5 μM paclitaxel were used in this study. We evaluated the effects of flubendazole and paclitaxel on the proliferation, speed, and total number of cells by 3D cell culture, cell colony formation, and EdU incorporation experiments. We found that flubendazole enhanced the inhibitory effect of paclitaxel on cell proliferation in MDA-MB-231 and MCF-7 cells (Figure 1C–N).

We then performed scratch and transwell experiments to verify the anti-metastatic potential of the combination therapy in MCF-7 and MDA-MB-231 cells. Following the administration of the combined treatment with flubendazole and paclitaxel, both the healing area (Figure 2A–D) and the migration rate of cells (Figure 2E–H) were reduced. Moreover, we detected changes in the fluorescence intensity of E-cadherin and matrix metalloproteinase 2 (MMP-2) by immunofluorescence under the intervention of flubendazole and paclitaxel. The outcomes showed that the combination of flubendazole and paclitaxel significantly enhanced E-cadherin’s fluorescence intensity (Figure 2I–N) and weakened MMP-2′s fluorescence intensity compared with the control (Figure 2K–P). Meanwhile, we also detected the expression of metastasis-related proteins in MDA-MB-231 and MCF-7 cells using Western blot. The results showed that the combination of flubendazole and paclitaxel significantly increased the protein expression level of E-cadherin, and significantly reduced the protein expression levels of MMP-2 and N-cadherin. (Figure 2Q,R). Overall, these findings indicated that flubendazole could enhance the inhibitory effect of paclitaxel in BC.

### 2.2. Combination of Flubendazole and Paclitaxel Leads to Aberrant Mitosis and Induces Apoptosis in BC Cells

To further understand the effect of the drug combinations on microtubule assembly, we conducted confocal microscopies in MDA-MB-231 and MCF-7 cells using antibodies against α-Tubulin and β-Tubulin. These data indicated that the combined use of flubendazole and paclitaxel could exert a significant impact on the stability of microtubules (Figure 3A–D). A flow cytometry analysis showed that after treatment of TNBC cells with flubendazole and paclitaxel, the G2/M phase was significantly prolonged to varying degrees in MDA-MB-231 cells and MCF-7 cells, and the proportion of the G2/M phase in the combined application group was increased compared to the flubendazole group and paclitaxel group in MCF-7 cells (Figure 3E–H), suggesting that the flubendazole and paclitaxel led to tumor cell cycle arrest, and the combined application group could induce stronger tumor cell cycle arrest in MCF-7 cells. In order to further detect the effect of flubendazole and paclitaxel on apoptosis in TNBC cells, we used annexin-V/PI double staining flow cytometry to obtain apoptosis signals at the molecular level for quantitative analysis of tumor cells. The results showed that the combination of flubendazole and paclitaxel significantly increased the apoptosis rate of cells compared with the control group, with a marked increase in the proportion of late apoptotic cells (Figure 3I–L). Altogether, our findings suggest that combining flubendazole and paclitaxel leads to aberrant mitosis and induces apoptosis in BC.

### 2.3. Combination of Flubendazole and Paclitaxel Regulates HIF1α/PI3K/AKT Signaling Pathways

To elucidate the underlying mechanism of flubendazole combined with paclitaxel in the therapy of BC, we performed RNA sequencing (RNA-seq) analyses on MDA-MB-231 cells treated with flubendazole and paclitaxel for 24 h. Compared with the control group, there were 1939 genes differentially expressed in the samples treated with flubendazole-paclitaxel, including 1001 up-regulated genes and 938 down-regulated genes (Figure 4A). Through KEGG enrichment analysis of the functional changes in differentially expressed genes, 12 signal pathways were found to have changed (Figure 4B). These signal pathways are involved in controlling many biological processes related to metabolic processes inside and outside cells. Among these signal pathways, PI3K/AKT is a classic signal pathway that responds to insulin, guides cells to absorb and utilize nutrients, and promotes metabolism, proliferation, cell survival, growth, and angiogenesis [18]. The PI3K/AKT signal pathway is often widely activated in tumor genesis [19]. After the completion of KEGG enrichment, in order to further analyze the expression of PI3K/AKT-related genes, the heat map was drawn by using the pheatmap package (R version 4.1.2) (Figure 4C). In order to analyze the interaction between these genes and proteins, we analyzed the protein interaction through STRING data, and drew the PPI network using the software of cystoscope (version 3.9.1) (Figure 4D). Through the analysis of the above biographical information, we found that there were 21 gene expression differences, including 6 up-regulated genes and 15 down-regulated genes. Among them, HIF1α emerges as a potential key molecule, and intriguingly, its expression was effectively reduced by treatment with flubendazole, paclitaxel, and their combination. And compared with monotherapy, the combination therapy shows high activity (Figure 4C,D). In MDA-MB-231 and MCF-7 cells, we analyzed the protein expression of HIF1α and classical PI3K/AKT signal pathway-related molecules before and after treatment with flubendazole, paclitaxel, and their combination. The combination of flubendazole and paclitaxel significantly decreased the expression of p-mTOR, p-AKT, p-PI3K, and HIF1α (Figure 4E,F). As a result, it is suggested that the combination of flubendazole and paclitaxel may play an anti-BC role by regulating the HIF1α/PI3K/AKT pathway.

### 2.4. Flubendazole Combined with Paclitaxel Suppresses Tumor Growth and Metastasis In Vivo

To assess the in vivo anti-tumor proliferation effect of the combination of flubendazole and paclitaxel, we inoculated MDA-MB-231-luc cells into the nude mice model by subcutaneous injection and finally screened the qualified xenograft tumor model in nude mice. The results showed that the combination of flubendazole and paclitaxel reduced the fluorescence intensity compared with the control group (Figure 5A,B). Furthermore, we discovered that the tumor size grew more slowly, the tumor volume was lower, and the tumor mass was lighter in the combined treatment group (Figure 5C–E). Upon immunohistochemical examination of tumor tissue, it was determined that the positive cell infiltration rate of proliferation-related factor Ki67 was substantially lower in the combined treatment group (Figure 5F,G). Additionally, the TUNEL chromogenic assay demonstrated that the fluorescence intensity of the combined treatment group was the highest (Figure 5H,I). To evaluate the cytotoxic effect of combined treatment on the normal tissue of the nude mice xenotransplantation tumor model, we monitored the weight of the animal models every day and discovered that the weight of each group of models increased steadily, and the combination treatment of the high-dose group had no significant effect on the weight of the animal model (Figure 5J). Finally, H&E staining on the heart, liver, spleen, lung, and kidney showed no apparent damage, suggesting that flubendazole and paclitaxel had no noticeable adverse reactions in the xenograft tumor model of BC nude mice (Figure 5K).

To verify the anti-metastasis activity of combined therapy in vivo, we inoculated MDA-MB-231-luc cells into the tail vein of nude mice. Analysis of the imaging data revealed the absence of distant metastasis in the kidney, liver, spleen, and heart (Figure 6A,B). Then, the level of metastasis-related proteins in lung tissues was detected by immunohistochemistry. The outcomes showed that combined treatment could increase the positive cell staining rate of E-cadherin (Figure 6C,D) and reduce the positive cell staining rate of MMP-2 and N-cadherin (Figure 6E–H). Collectively, flubendazole combined with paclitaxel suppresses tumor growth and metastasis.

### 2.5. Flubendazole Inhibits the Proliferation and Migration of MCF-7/PTX Cells

To verify whether flubendazole can overcome the resistance of BC to paclitaxel, we detected the effects of different concentrations of flubendazole on cell proliferation and metastasis in MCF-7/PTX cells. The results of a 3D cell culture test and colony formation test revealed that after treatment with flubendazole, the size of MCF-7/PTX 3D tumor cell spheres decreased (Figure 7A,B) and the proliferation rate of tumor cells slowed down (Figure 7C,D). The transwell assay showed that after treatment with flubendazole, the number of tumor cells transferred to MCF-7/PTX decreased (Figure 7E,F). Moreover, we found that flubendazole can reduce the expression of HIF1α, p-PI3K, p-AKT, and p-mTOR in drug-resistant cells by analyzing the Western blot results (Figure 7G,H). These findings collectively indicate that treatment with flubendazole in MCF-7/PTX cells leads to varying degrees of inhibition in the proliferation and metastasis of BC cells, accompanied by diverse reductions in the expression of the PI3K/AKT signaling pathway. Moreover, flubendazole promoted the degradation of p62 and the transcription of LC3 I to LC3 II in MCF-7/PTX cells, indicating that flubendazole could induce autophagy. These outcomes indicate that flubendazole has the potential to overcome drug resistance in BC cells and restore their sensitivity to paclitaxel.

## 3. Discussion

As a member of the taxane family, paclitaxel is used as a classic anticancer chemotherapy drug by stabilizing microtubules and is approved by the FDA for use in BC [20]. Paclitaxel can attach to microtubules and stabilize them by encouraging the assembly of α-and β-tubulin subunits. It interferes with microtubule dynamics by reducing the shortening of Tubulin polymer, resulting in the destruction of the cell’s ability to divide those arrests at G2/M phase and cell apoptosis [21,22]. Unfortunately, most BC patients have inherited or acquired resistance to paclitaxel treatments and relapsed [23]. In clinical practice, combination therapy has been suggested as a means of preventing and overcoming drug resistance, which aims to improve efficacy through drug synergy and to lessen the frequency and severity of side effects by reduced doses of individual drugs [24,25]. Several drugs have been reported to combinate with paclitaxel in cancer [26], such as atezolizumab [27,28] and resveratrol [29]. In addition, some studies have found that flubendazole can enhance the cytotoxicity of fluorouracil and adriamycin by inhibiting tubulin polymerization and inducing the formation of monopolar spindle in breast cancer [30]. Flubendazole as a benzimidazole derivative has been available for human use for the treatment of gastrointestinal nematode [31]. Current preclinical researches have shown that flubendazole has inhibitory effects on diverse types of tumors [32,33], including leukemia, myeloma, neuroblastoma, oral squamous carcinoma, as well as liver cancer, colon cancer, and BC [9,13,14,15,34,35,36,37]. In addition, flubendazole has been recognized as a microtubule inhibitor that can disrupt microtubule polymerization by binding to free tubulin dimers at the colchicine binding site [38,39,40], whereas paclitaxel inhibits the microtubule depolymerisation by binding to tubulin monomers at another diverse site [41]. Interestingly, orally inhalable flubendazole nanocrystals also synergistically enhance the cytotoxicity of paclitaxel on A549 lung cancer cells, reduce adverse reactions, and solve the problems of poor solubility and bioavailability of flubendazole in clinical practice [42]. Therefore, the combination of flubendazole and paclitaxel is expected to have largely additive anti-tumor effects. Consistent with previous findings, this study demonstrates that the combination of flubendazole and paclitaxel can have a synergistic effect on BC cells. In this regard, we demonstrate that combination therapy has a remarkable inhibitory effect on BC, associated with a significant negative impact on microtubule dynamics. This study supports that flubendazole can be an effective synergist for paclitaxel and the synergistic combination of flubendazole with low-dose paclitaxel could be developed as an effective therapeutic modality.

To further illustrate the mechanisms of action of flubendazole and paclitaxel combination treatment, we used an RNA-seq analysis and KEGG enrichment analysis to check the changes in the content and function of differentially expressed genes. The results found that 12 signaling pathways had changed, which are involved in controlling many biological processes related to metabolic processes inside and outside cells. Among these signaling pathways, PI3K/AKT is a classic signaling pathway that responds to insulin, guides cells to absorb and utilize nutrients, and promotes metabolism, proliferation, cell survival, growth, and angiogenesis. By remodeling BC cell metabolism, AKT plays a crucial regulatory function in the drug-resistance mechanism of BC. Specifically, inhibiting AKT can overcome the drug resistance of immunotherapy to chemotherapy in BC cells [43]. In PI3K/AKT-resistant TNBC cells, chloroquine (CQ) has been shown to enhance the synergistic effect between paclitaxel-based chemotherapy and PI3K/AKT inhibitors by modulating autophagy to counteract drug resistance [44]. Therefore, we focused on exploring the mechanisms of the PI3K/AKT signaling pathway in the combination of paclitaxel and flubendazole. We used the heatmap package, STRING database, and Cytoscape to analyze PI3K/AKT-related genes, and there were 21 gene expression differences, including 6 upregulated genes and 15 downregulated genes. Among these genes, it is reported that HIF1 is increased in tumor tissue. HIF1 is made up of two subunits, HIF1α and HIF1β, each with its own specialized function; HIF1α is required for gene expression in hypoxic cells [45]. Multiple studies have indicated that HIF1α expression is elevated in numerous malignancies and promotes cancer progression [46]. The RNA-seq results indicate that the use of paclitaxel or flubendazole or a combination lowers the expression of HIF1α in BC cells. It has been reported that 2-methoxyestradiol downregulates HIF1α through binding to tubulin and blocking tumor interphase microtubules that are needed for HIF1α down-regulation [47]. Mechanistically, as microtubule inhibitors, paclitaxel and flubendazole may inhibit HIF1α by interfering microtubule dynamics. The combination of flubendazole and paclitaxel significantly decreased the expression of HIF1α, p-PI3K, p-AKT, and p-mTOR. These findings suggest that the combination of flubendazole and paclitaxel could affect microtubule dynamics, down-regulate HIF1α, and inhibit the PI3K/AKT signaling pathway, which results in synergistic anti-tumor effects for BC therapy.

More importantly, recent research demonstrated that flubendazole has the potential to overcome trastuzumab resistance in BC [11], which makes us interested in the ability of flubendazole to reverse paclitaxel resistance. Our results showed that flubendazole treatment inhibited proliferation and metastasis in both paclitaxel-sensitive and paclitaxel-resistant BC lines, which indicated the potential of flubendazole to overcome resistance to paclitaxel treatment for BC therapy. Hyperactivation of the PI3K/AKT signaling pathway is related to tumor genesis, metastasis, and chemotherapy resistance [48]. AKT functions as a key regulatory in drug resistance by remodeling BC cell metabolism, and inhibiting AKT can resist the drug resistance of immunotherapy to chemotherapy in BC cells [43]. A study unveiled that macrophage-capping proteins (CapG) facilitate the drug resistance of BC to paclitaxel by mediating the PI3K/AKT signaling pathway, accompanied by a poor prognosis of BC patients [49]. Significantly, HIF1α contributes to the development of drug resistance to radiotherapy and chemotherapy in cancer treatment [50,51]. In paclitaxel-resistant BC cells (MCF-7/PTX cells), flubendazole administration suppressed the PI3K/AKT pathway and decreased the expression of HIF1α. These results establish flubendazole as a repositioning chemosensitizer of paclitaxel and highlight the anti-tumor mechanisms of the combination of flubendazole and paclitaxel. Moreover, since HIF1α participates in many other pathways except the PI3K/AKT pathway, such as the MAPK/ERK and JAC/STAT3 pathways [52], we believe that further research is required to figure out other mechanisms related to the role of HIF1α in paclitaxel resistance. Interestingly, flubendazole inhibited the proliferation of various types of cancer cells through regulating autophagy [53,54,55]. In addition, our previous research found that flubendazole could treat breast cancer by regulating autophagy [13,14,15]. In this study, we also found that flubendazole can increase the expression level of autophagy-related factors and induce autophagy in MCF-7/PTX cells. It is suggested that flubendazole might induce autophagy to treat breast cancer. However, it is necessary to investigate the mechanisms behind future investigations.

Our findings reveal the novel mechanisms of the synergetic anti-tumor effects of a paclitaxel and flubendazole combination on BC. The present study provides convincing data to establish an enhanced anti-tumor effect of paclitaxel caused by combining flubendazole via HIF1α/PI3K/AKT signaling pathways for BC therapy, which contributes to reducing the clinical dose of paclitaxel, thus diminishing baneful adverse reactions. Moreover, this study provides the first evidence that combining flubendazole treatment can be a potential therapeutic strategy in paclitaxel-resistant BC, along with results that flubendazole could repress the growth and metastasis of paclitaxel-resistant BC cells. Taken together, these findings highlight the synergistic anti-tumor effects of combining paclitaxel with flubendazole, representing a valuable approach to enhance therapeutic efficacy and overcome resistance to paclitaxel monotherapy in BC.

## 4. Materials and Methods

### 4.1. Cell Culture and Reagents

MCF-7 and MDA-MB-231 originated from ATCC and were maintained in Dulbecco’s Modified Eagle Medium (DMEM) containing 10% fetal bovine serum (FBS) and 1% penicillin–streptomycin (Life Technologies, Carlsbad, CA, USA) at 37 °C in 5% CO_2_. Reagents used in this study were as follows: MTT (M2128), DAPI (D9542), Flubendazole (HY-B0294), Paclitaxel (HY-B0015), MMP-2 (87809, CST), E-cadherin (14472, CST), β-actin (66009-1-Ig, Proteintech, Rosemont, IL, USA), N-Cadherin (13116, CST), PI3K (4255, CST), p-PI3K (17366, CST), AKT (9272, CST), p-AKT (4060S, CST), mTOR (2972, CST), p-mTOR (5536, CST), HIF1α (36169, CST) p62 (88588, CST), LC3 (4108, CST), ATG5 (12994, CST).

### 4.2. Cell Viability Assay

The MDA-MB-231 and MCF-7 cells were subjected to treatment with either paclitaxel or flubendazole in accordance with the experimental design. Subsequently, the intervention was concluded. The cell culture medium was discarded and an appropriate volume of incomplete culture medium was withdrawn. The necessary dosage of MTT was determined at 20 μL per well. To dissolve the purple crystals, 150 μL of DMSO was added to each well, followed by gentle agitation in the dark for 10 min. The absorbance of each well was then measured at a wavelength of 490 nm using a microplate spectrophotometer (EON, Bio-Tek, Winooski, VT, USA), and the resulting measurements underwent statistical analysis.

### 4.3. Combined Drug Effect Analysis

CompuSyn software was used to produce a normalized equivalent line diagram and fractional impact composite index (CI) diagram [56]. CI value < 1 is synergistic effect, =1 is additive effect, and > 1 is antagonistic effect.

### 4.4. Colony Formation Assay

We inoculated 1000 cells per well into a 6-well plate and placed them in a cell incubator. After overnight incubation when the cells demonstrated visible proliferation, the intervention was concluded according to the experimental design following treatment with the compounds [57]. The culture medium was discarded and the cells fixed with 4% paraformaldehyde. After dyeing with 0.5% crystal violet solution for 1 h, we observed the cells under an inverted microscope.

### 4.5. Three-Dimensional Cell Culture

We inoculated cells in a 3D cell culture plate (S-bio; catalog number MS9096UZ, Hudson, NH, USA) and placed in a cell incubator to form uniform and dense 3D cell microspheres. Then, the cells were stained with Hoechst 33342 (Beyotime; Cat: C1022, Shanghai, China) and photographed with a fluorescence microscope [57].

### 4.6. Immunofluorescence

The logarithmic growth phase cells were taken and inoculated with 2.5x10^4^ cells per well on a 24-well plate. After 24 h, according to the experimental design, they were treated with the compounds and then the intervention was terminated. Then, 4% paraformaldehyde was fixed for 15 min and 5% BSA was closed for 1 h. The blocking solution was discarded and primary antibodies and secondary antibodies (TRITC, ab6718; FITC, ab6717) were added to each well. The coverslip was processed with an anti-fluorescence quenching solution, excess liquid removed with a cotton swab, and the samples were directly observed using a fluorescence microscope [57].

### 4.7. Western Blotting

The treated MDA-MB-231 and MCF-7 cells were washed twice with 1 mL pre-cooled PBS and centrifuged at 3500 rpm for 5 min. The residual PBS was absorbed, 200 μL of lysate added, and the cells were broken with maximum power ultrasound in an ice bath. The lysate was then subjected to centrifugation at 12,000 rpm at 4 °C for 15 min, and the resulting supernatant was collected. The protein concentration was determined using the BCA method. Based on the quantitative protein results, the total protein samples were mixed with 5× protein loading buffer and loaded into the gel wells. Electrophoresis was conducted at a voltage of 80 V and transfer membrane for 1 h was added with the first antibody overnight and added with the second antibody for 2 h. Finally, the chemiluminescence imaging system was used for imaging and photographing.

### 4.8. Annexin-V/PI Dual Staining

The treated cells in each group were centrifuged at 1500 rpm to pellet the cells and carefully remove the supernatant. Pre-cooled 1× binding buffer (100 μL) was added, followed by sequential addition of propidium iodide and annexin V-FITC staining solution. It was mixed thoroughly and incubated in the dark at room temperature for 15 min. Afterward, 400 μL 1× binding buffer was added and the percentage of apoptotic cells was assessed using flow cytometry [13].

### 4.9. Scratch Assay

The treated cells were seeded at a density of 3 × 10^5^ cells per well and incubated overnight. The following day, a scratch was created perpendicular to the horizontal axis using a pipette tip. Cell samples were collected at 0 and 24 h and images were captured using a phase-contrast microscope. [13].

### 4.10. Transwell Migration Assay

Prior to the preparation of the cell suspension, cells were subjected to serum deprivation in serum-free medium. Subsequently, 100 µL of the cell suspension was added to a Transwell cell insert (8 µm pore size, Millipore, Burlington, MA, USA). After removing the Transwell cell, the culture solution in the insert was discarded and the insert was washed twice with PBS. It was then fixed with methanol for 30 min and allowed to air dry. Subsequently, the cells were stained with crystal violet and observations were made under a microscope [13].

### 4.11. Cell Cycle Distribution

The cells were treated with compounds for 24 h. Following treatment, they were fixed overnight in 4 °C absolute ethanol. The next day, the cell cycle was analyzed using a cell cycle detection kit where propidium iodide was incubated with the cells, and cell cycle analysis was performed using flow cytometry [13].

### 4.12. Animal Experiments

Subcutaneous Xenograft Model: MDA-MB-231-luc cells (5 × 10^6^) were subcutaneously implanted into 24 female nude BALB/c mice (6-8 weeks, weighing 20–22 g). Approximately one week later when the tumors reached a volume of 100 mm3 (calculated as V = L × W2/2), the mice were divided into four groups: a control group (untreated), a flubendazole group (15 mg/kg/day), a paclitaxel group (5 mg/kg/day), and a combination group. Intravenous Xenograft Model: MDA-MB-231-Luc cells (5 × 10^6^) were intravenously implanted into 24 female nude BALB/c mice (6–8 weeks old, weighing 20–22 g). Mouse weight and tumor size were recorded daily during the course of treatment. The bioluminescence of the subcutaneous tumors in the mice was visualized using IVIS Spectrum.

### 4.13. Immunohistochemistry

The tissue sections were dewaxed and rehydrated to inhibit endogenous peroxidase activity. After washing and antigen repair, the sections were sealed in non-immune serum, then incubated with the first antibody, followed by the second antibody and colored substrate solution. Finally, the sections were re stained, washed, dehydrated, and sealed with hematoxylin [13].

### 4.14. TUNEL Assays

Apoptotic cells were detected using the terminal deoxynucleotidyl transferase-mediated deoxyuridine triphosphate (dUTP) nick end labeling (TUNEL) method. Tissue samples were embedded in paraffin and cut into 4 μm sections. These sections were then deparaffinized in xylene and dehydrated with a graded alcohol series. Protease K (20 mg/L) digestion was performed at 37 °C for 30 min. Next, a reaction mixture containing terminal deoxynucleotidyl transferase and fluorescein-labeled dUTP was added and the samples were incubated at 37 °C for 2 h. Anti-fluorescein antibodies conjugated with alkaline phosphatase were then applied dropwise and incubated for 30 min at 37 °C. Hematoxylin was used for light contrast staining, followed by standard dehydration, clearing, sealing, and observation under a microscope. [13].

### 4.15. H&E Staining

The tissue samples from xenotransplantation model mice, which were fixed in 4% paraformaldehyde, were embedded in paraffin and subsequently sectioned. These sections underwent dewaxing with xylene and dehydration using a gradient ethanol solution. Hematoxylin staining was performed for a duration ranging from 2 to 5 min, depending on room temperature conditions. The sections were then subjected to differentiation in a differentiation solution for 10–30 s. After differentiation, they were dehydrated using a gradient ethanol solution and cleared with xylene twice, each time for one minute. Finally, the sections were removed from the xylene and sealed with neutral gum [13].

### 4.16. Transcriptome Analysis

We employed R software (version 4.1.2) to visualize data using the “plot” package (R version 4.1.2). To delve into the functional analysis of KEGG enrichment for differentially expressed genes, we utilized the “clusterProfiler” package (R version 4.1.2) for enrichment and then employed the “plot” package to visualize the results of the enrichment analysis. Upon the completion of KEGG enrichment, we further scrutinized the expression of genes related to the PI3K/AKT pathway. To examine the interactions between these gene products, we conducted protein–protein interaction (PPI) analysis using STRING data (https://cn.string-db.org/; accessed on 1 January 2023). The PPI network was generated and visualized using Cytoscape software (version 3.7.2) [58].

### 4.17. Statistical Analysis

Statistical analysis was conducted using SPSS software version 25.0 and GraphPad Prism 8.0 software. The mean values of independent sample groups were compared using a *t*-test. Differences among multiple groups were analyzed through one-way ANOVA, followed by Dunnett’s post hoc test. A significance level of *p* < 0.05 was considered as indicative of statistical significance.

## 5. Conclusions

In summary, our results demonstrate that flubendazole enhances the inhibitory effect of paclitaxel via the HIF1α/PI3K/AKT signaling pathways, offering valuable insights for finding more novel combination therapies for BC in the future.

## Figures and Tables

**Figure 1 ijms-24-15121-f001:**
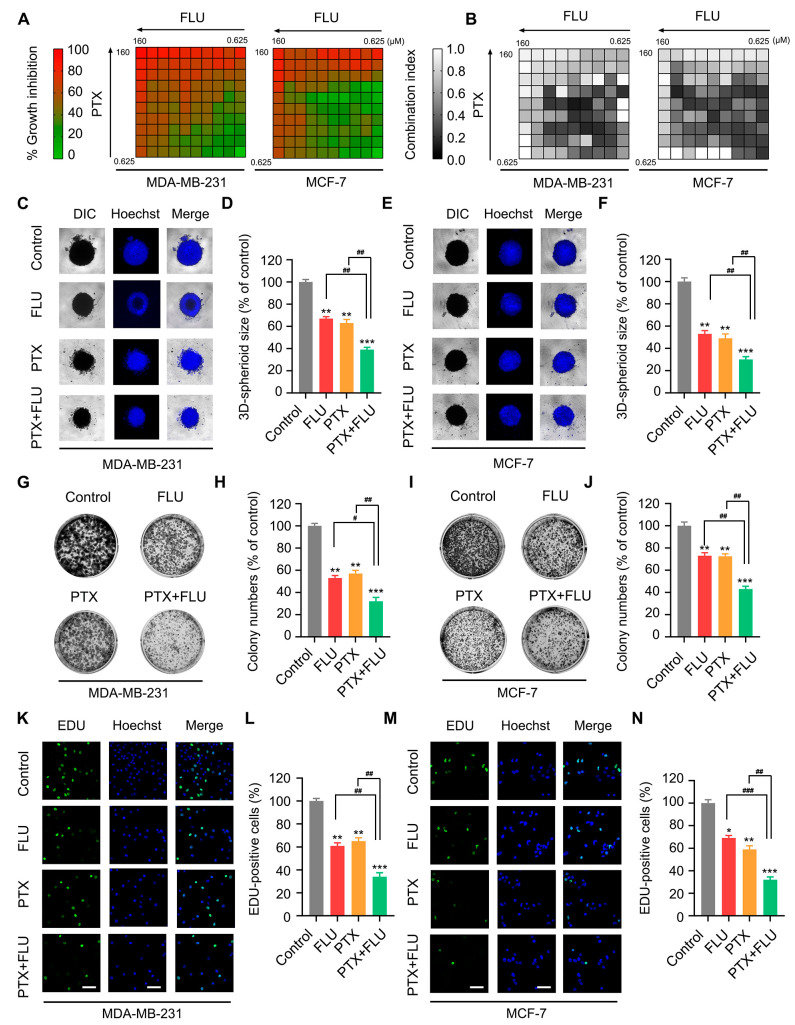
Flubendazole enhances the proliferation inhibitory effect of PTX in MDA-MB-231 and MCF-7 cells. (**A**,**B**) heat map of mean IC_50_ values of flubendazole and paclitaxel in MDA-MB-231 and MCF-7 cell lines at different concentrations. Cell viabilities were measured by MTT assay. Combination indexes were calculated by compusyn software (https://www.combosyn.com/index.html accessed on 28 September 2023). (**C**–**F**) 3D cell culture assay of MDA-MB-231 and MCF-7 cells treated with vehicle, flubendazole, paclitaxel, flubendazole combined with paclitaxel for 24 h. Representative images and quantitative analysis of 3D cell are shown. (**G**–**J**) colony formation assay of MDA-MB-231 and MCF-7 cells treated with vehicle, flubendazole, paclitaxel, flubendazole combined with paclitaxel for 24 h. Representative images and quantitative analysis of colonies are shown. (**K**–**N**) EdU incorporation assay of MDA-MB-231 and MCF-7 cells treated with vehicle, flubendazole, paclitaxel, flubendazole combined with paclitaxel for 24 h. Representative images and quantitative analysis are shown. Scale bar,100 µm. Data are expressed as mean ± SEM. All data are representative of at least three independent experiments. * *p* < 0.05, ** *p* < 0.01, *** *p* < 0.001, compared with control; ^#^
*p* < 0.05, ^##^
*p* < 0.01, ^###^
*p* < 0.001 compared with the flubendazole and paclitaxel combinational group.

**Figure 2 ijms-24-15121-f002:**
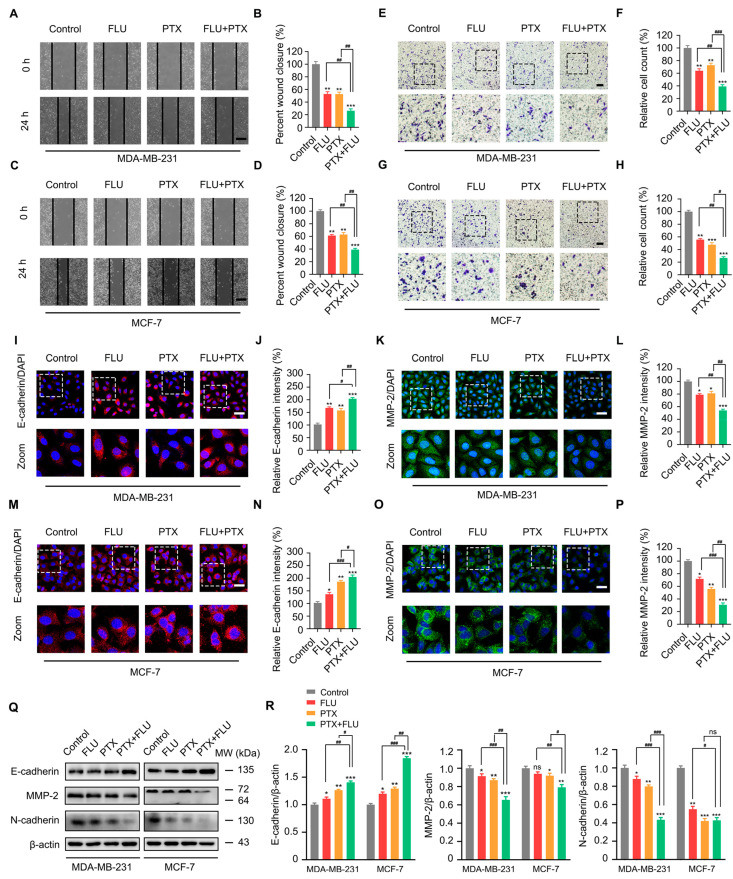
Flubendazole enhances the metastasis inhibitory effect of PTX in MDA-MB-231 and MCF-7 cells. (**A**–**D**) scratch assay of MDA-MB-231 and MCF-7 cells treated with vehicle, flubendazole, paclitaxel, flubendazole combined with paclitaxel for 24 h. Representative images and quantitative analysis of scratch cells are shown. Scale bar,100 µm. (**E**–**H**) transwell assay of MDA-MB-231 and MCF-7 cells treated with vehicle, flubendazole, paclitaxel, flubendazole combined with paclitaxel for 24 h. Representative images and quantitative analysis of percentage of positive ratios are shown. Scale bar, 40 µm. (**I**–**P**) immunofluorescence analysis of the E-cadherin and MMP-2 antibody in MDA-MB-231 and MCF-7 cells treated with vehicle, flubendazole, paclitaxel, flubendazole combined with paclitaxel for 24 h. Representative images with quantification of E-cadherin and MMP-2 intensity are shown. Scale bar, 20 µm. (**Q**–**R**) Western blotting analysis of E-cadherin, MMP-2, and N-cadherin expression in MDA-MB-231 and MCF-7 cells treated with vehicle, flubendazole, paclitaxel, flubendazole combined with paclitaxel for 24 h. β-actin was used as a loading control. Quantification of Western blotting analysis is shown. Data are expressed as mean ± SEM. All data are representative of at least three independent experiments. * *p* < 0.05, ** *p* < 0.01, *** *p* < 0.001, compared with control; ^#^
*p* < 0.05, ^##^
*p* < 0.01, ^###^
*p* < 0.001 compared with the flubendazole and paclitaxel combinational group; ns, no significance.

**Figure 3 ijms-24-15121-f003:**
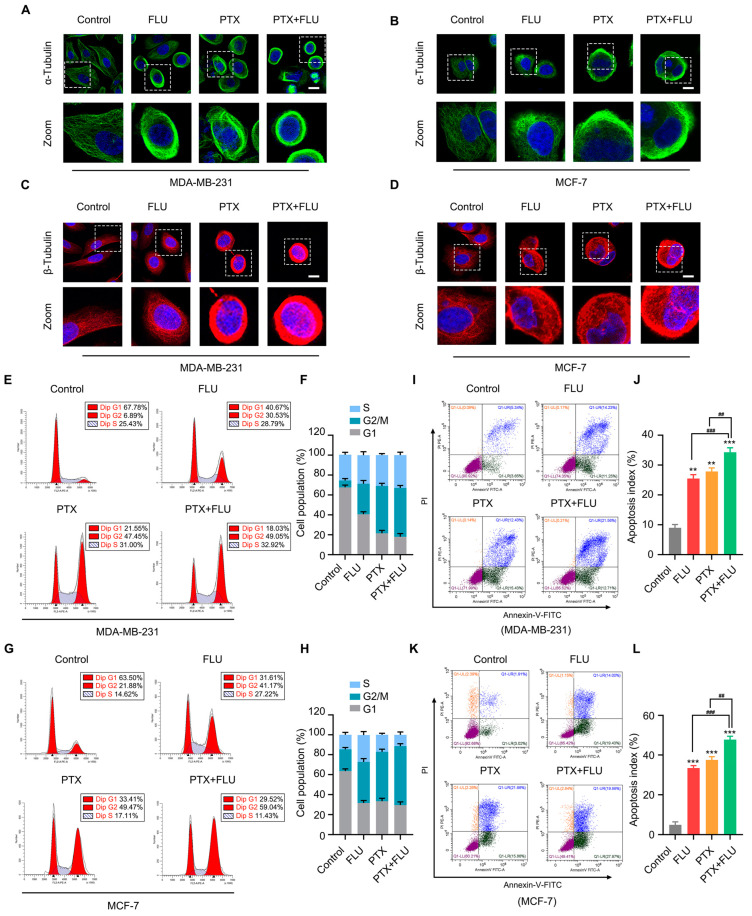
Combination of flubendazole and paclitaxel leads to aberrant mitosis and induces apoptosis in MDA-MB-231 and MCF-7 cells. (**A**–**D**) immunofluorescence image of α-Tubulin and β-Tubulin antibodies in MDA-MB-231 and MCF-7 cells after treatment by vehicle, flubendazole, paclitaxel, flubendazole combined with paclitaxel. Scale bar, 10 µm. (**E**–**H**) cell cycle distribution assay of MDA-MB-231 and MCF-7 cells treated with vehicle, flubendazole, paclitaxel, flubendazole combined with paclitaxel. Representative images and quantitative analysis of the cell population are shown. (**I**–**L**) MDA-MB-231 and MCF-7 cells were treated with vehicle, flubendazole, paclitaxel, flubendazole combined with paclitaxel for 24 h, apoptosis ratios were determined by flow cytometry analysis of Annexin-V/PI double staining. Representative images and quantification of apoptosis are shown. Data are expressed as mean ± SEM. All data are representative of at least three independent experiments. ** *p* < 0.01, *** *p* < 0.001, compared with control; ^##^
*p* < 0.01, ^###^
*p* < 0.001 compared with the flubendazole and paclitaxel combinational group.

**Figure 4 ijms-24-15121-f004:**
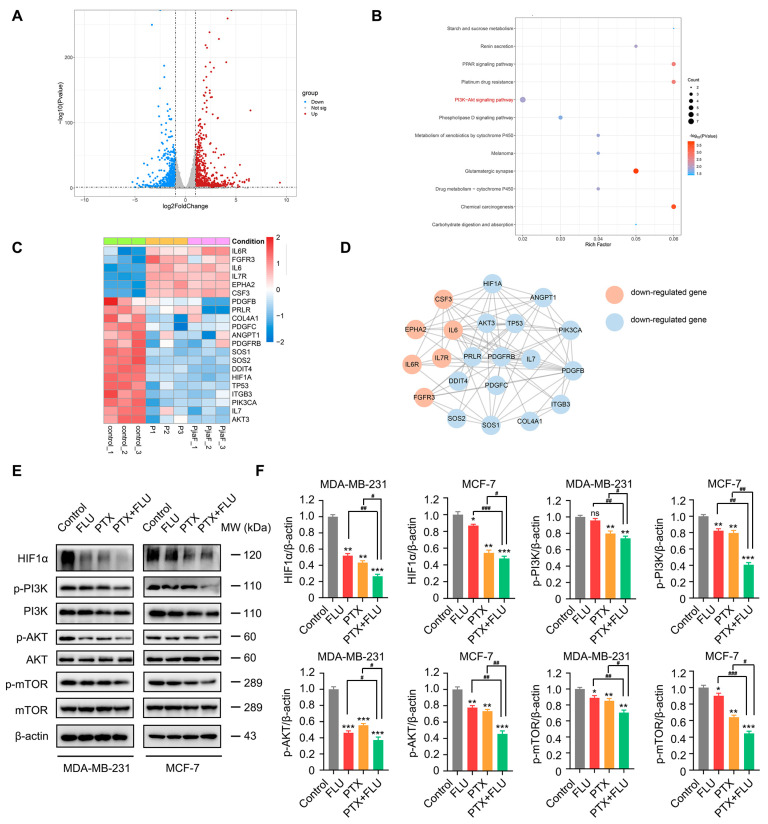
Transcriptome analysis of MDA-MB-231 cells treated with vehicle, paclitaxel, flubendazole combined with paclitaxel for 24 h. (**A**) volcanic map of significantly differentially expressed genes in paclitaxel group and flubendazole combined with paclitaxel group. (Red dots indicate genes with FDR < 0.05 and log2FoldChange > 1; Blue dots indicate genes with FDR < 0.05 and log2FoldChange < −1. (**B**) pathway bubble diagram of differentially expressed gene enrichment in paclitaxel group and flubendazole combined with paclitaxel group. (**C**) PI3K/AKT signaling pathway-related genes differentially expressed heat map. (**D**) HIF1α and AKT genes interact with some genes of PI3K/AKT signaling pathway. (**E**,**F**) Western blotting analysis of HIF1α, PI3K, p-PI3K, AKT, p-AKT, mTOR, and p-mTOR expression in MDA-MB-231 and MCF-7 cells treated with vehicle, flubendazole, paclitaxel, flubendazole combined with paclitaxel for 24 h. β-actin was used as a loading control. Quantification of Western blotting analysis is shown. Data are expressed as mean ± SEM. All data are representative of at least three independent experiments. * *p* < 0.05, ** *p* < 0.01, *** *p* < 0.001, compared with control; ^#^
*p* < 0.05, ^##^
*p* < 0.01, ^###^
*p* < 0.001 compared with the flubendazole and paclitaxel combinational group; ns, no significance.

**Figure 5 ijms-24-15121-f005:**
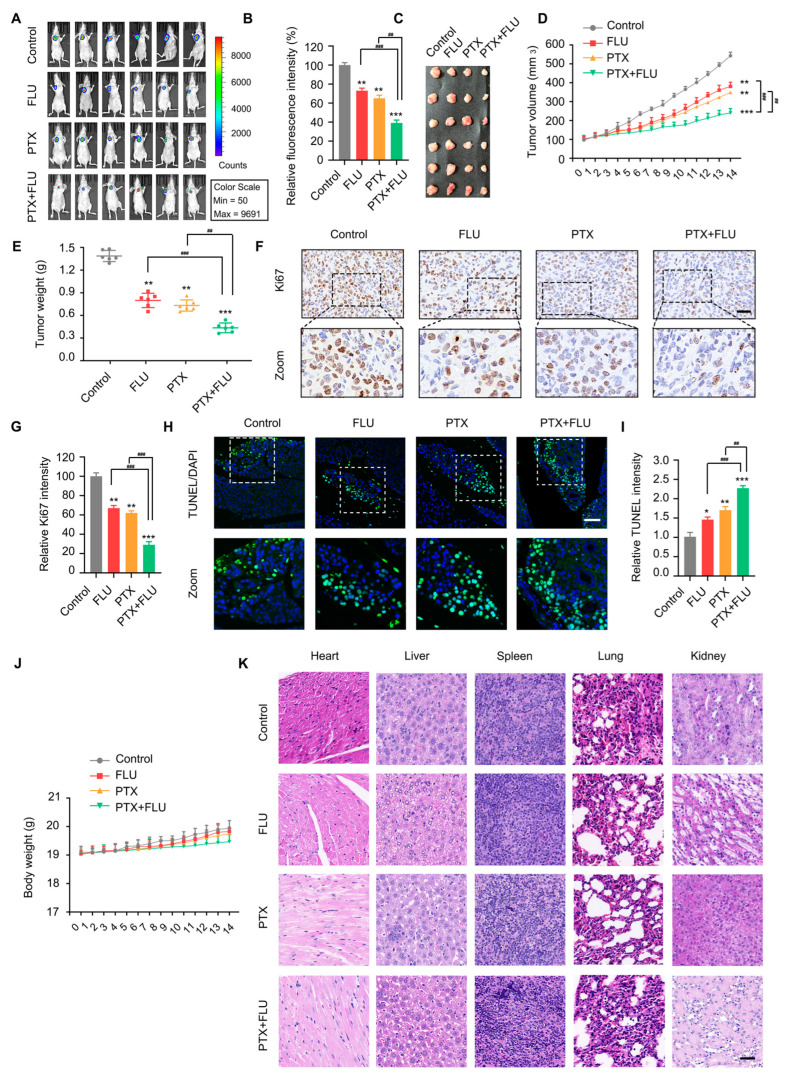
Flubendazole combined with paclitaxel suppresses tumor growth in vivo. (**A**,**B**) whole-body optical imaging system of subcutaneous breast cancer animal model of nude mice formed by injecting MDA-MB-231-luc cells after vehicle, flubendazole, paclitaxel, flubendazole combined with paclitaxel treatment for 24 h. Color scale, min = 50, max = 9691. (**C**) the images of isolated tumors derived from nude mice after vehicle, flubendazole, paclitaxel, flubendazole combined with paclitaxel treatment. (**D**,**E**) line graph of volume and weight of isolated tumors from nude mice with time. (**F**,**G**) the expression of Ki67 in MDA-MB-231 tumor sections of nude mice after vehicle, flubendazole, paclitaxel, flubendazole combined with paclitaxel treatment. Representative images and quantitative analysis of the percentage of positive ratios are shown. Scale bar, 40 µm. (**H**,**I**) TUNEL assay in tumor sections of nude mice after vehicle, flubendazole, paclitaxel, flubendazole combined with paclitaxel treatment. Representative images and quantification of TUNEL-positive cells are shown. Scale bar, 40 µm. (**J**) line graph of weight of nude mice with time. (**K**) H&E staining images representing the heart, liver, spleen, lung, and kidney of nude mice in each group are shown. Scale bar, 40 µm. Data are expressed as mean ± SEM. All data are representative of at least three independent experiments. * *p* < 0.05, ** *p* < 0.01, *** *p* < 0.001, compared with control; ^##^
*p* < 0.01, ^###^
*p* < 0.001 compared with the flubendazole and paclitaxel combinational group.

**Figure 6 ijms-24-15121-f006:**
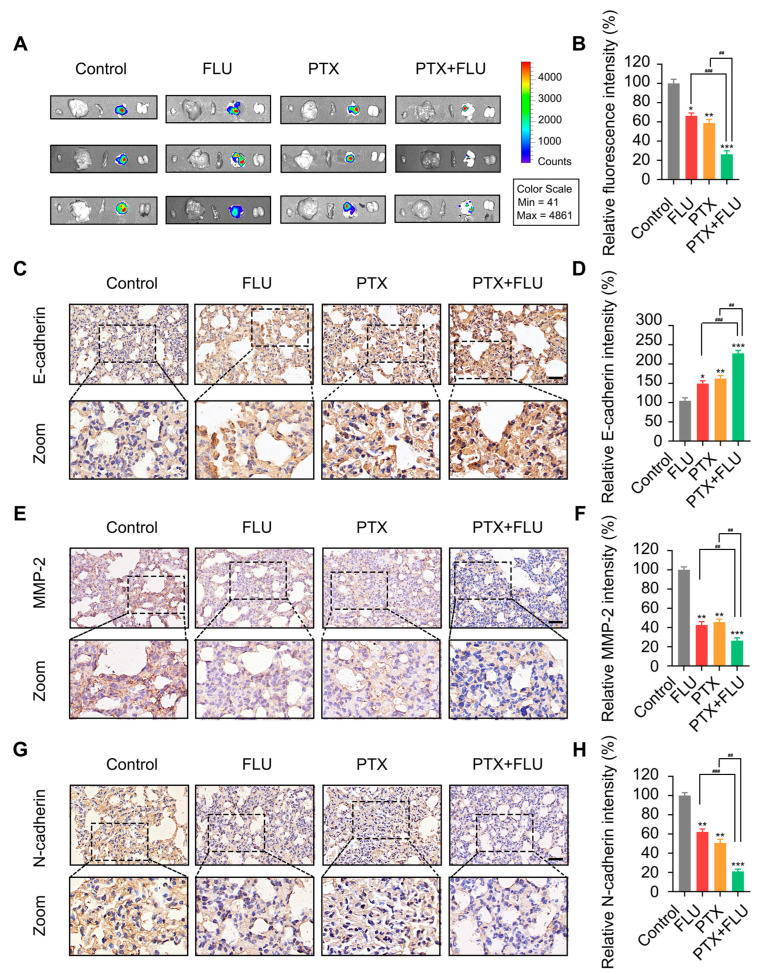
Flubendazole combined with paclitaxel suppresses tumor metastasis in vivo. (**A**,**B**) optical imaging system of heart, liver, spleen, lung, and kidney isolated from breast cancer animal model formed by injecting MDA-MB-231-luc cells into the vein of nude mice. Color scale, min = 41, max = 4861. Representative images and quantitative analysis of bioluminescent in lung sections are shown. (**C**–**H**) the expression of E-cadherin, MMP-2, and N-cadherin in lung sections of nude mice. Representative images and quantitative analysis of the percentage of positive ratios are shown. Scale bar, 40 µm. Data are expressed as mean ± SEM. All data are representative of at least three independent experiments. * *p* < 0.05, ** *p* < 0.01, *** *p* < 0.001, compared with control; ^##^
*p* < 0.01, ^###^
*p* < 0.001 compared with the flubendazole and paclitaxel combinational group.

**Figure 7 ijms-24-15121-f007:**
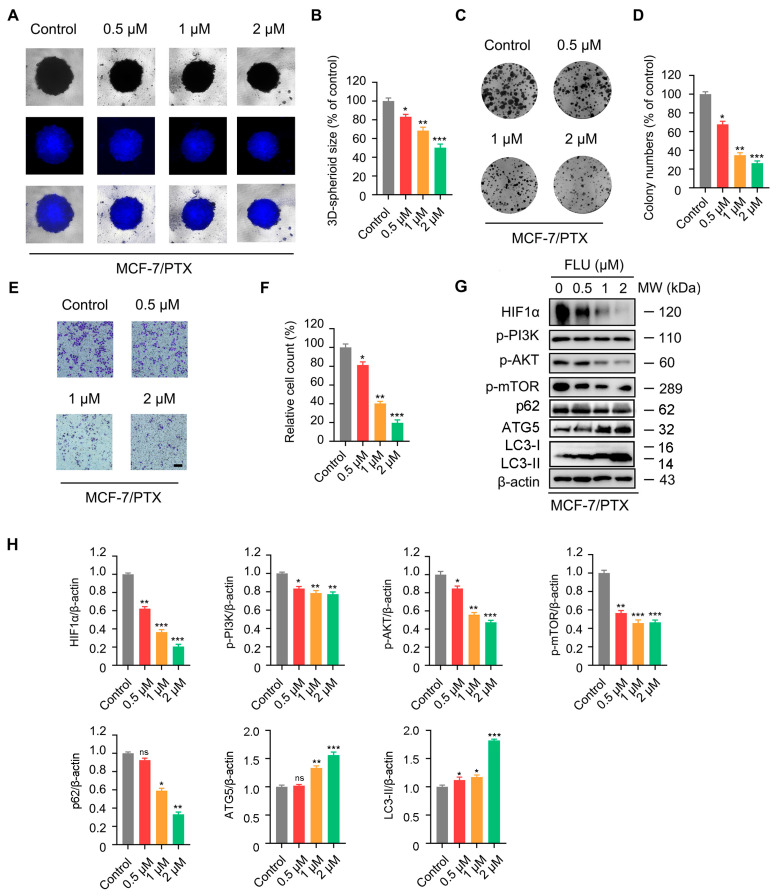
Flubendazole inhibits the proliferation and migration of paclitaxel-resistant MCF-7 cells. (**A**,**B**) 3D cell culture assay of MCF-7/PTX cells treated with vehicle and flubendazole. Representative images and quantitative analysis of 3D cells are shown. (**C**,**D**) colony formation assay of MCF-7/PTX cells treated with vehicle and flubendazole. Representative images and quantitative analysis of colonies are shown. (**E**,**F**) transwell assay of MCF-7/PTX cells treated with vehicle and flubendazole for 24 h. Representative images and quantitative analysis of percentage of positive ratios are shown. Scale bar, 40 µm. (**G**,**H**) Western blotting analysis of HIF1α, p-PI3K, p-AKT, p-mTOR, p62, ATG5, and LC3 expression in MCF-7/PTX cells treated with vehicle and flubendazole for 24 h. β-actin was used as a loading control. Quantification of Western blotting analysis is shown. Data are expressed as mean ± SEM. All data are representative of at least three independent experiments. Compared with the control group, * *p* < 0.05, ** *p* < 0.01, *** *p* < 0.001, ns, no significance.

## Data Availability

The data presented in the study are available in the article and the Appendix A. For further inquiries, you can directly contact the corresponding author.

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
