# Peer review of "Flubendazole Enhances the Inhibitory Effect of Paclitaxel via HIF1α/PI3K/AKT Signaling Pathways in Breast Cancer"

_ijms, 2023, doi:10.3390/ijms242015121_

Round 1

Reviewer 1 Report

The authors undertook to assess the effect of flubendazole and paclitaxel combination therapy on breast cancer cells. In vitro and in vivo experiments were used, which deserves recognition. I believe that the experiments are planned and performed correctly, however, I have a few comments regarding the submitted manuscript:

1. In the 4. line after the word "and" the name of author is missing.

2. I propose to add information about the role of the PI3K/AKT signaling pathway in breast cancer in the introduction of the paper.

3. In the materials and methods, please add information about the culture conditions used in the cell line experiments.

4. There is no information in the materials and methods regarding cell density in the suspension administered to experimental animals.

5. Were the same concentration of chemotherapeutic agents used in the in vivo study as in the in vitro study? How long were the animals given drugs? Please complete this data in the manuscript.

6. In "in vitro studies", I suggest adding the results of examining the effect of the tested combination of chemotherapeutic agents on normal cells, e.g. fibroblasts. The presented experiments are well planned, however, their weakness is the lack of reference to experiments determining cytotoxicity on normal cells. If there are no such results, we do not know whether the combination of drugs used is not more toxic to normal cells of the body. I know that the effects of drugs on normal cells have been evaluated in vivo, but I believe that such in vitro data should also be included in the manuscript.

Author Response

Reviewer 1

The authors undertook to assess the effect of flubendazole and paclitaxel combination therapy on breast cancer cells. In vitro and in vivo experiments were used, which deserves recognition. I believe that the experiments are planned and performed correctly, however, I have a few comments regarding the submitted manuscript:

  1. In the 4. line after the word "and" the name of author is missing.

Author reply: Thank you for your comment. We have added in the revised manuscript.

  1. I propose to add information about the role of the PI3K/AKT signaling pathway in breast cancer in the introduction of the paper.

Author reply: Thank you for your suggestions. We have added information about the role of the PI3K/AKT signaling pathway in breast cancer in the revised manuscript. Such as “chloroquine (CQ) has been shown to enhance the synergistic effect between paclitaxel-based chemotherapy and PI3K/AKT inhibitors by modulating autophagy to counteract drug resistance.”

  1. In the materials and methods, please add information about the culture conditions used in the cell line experiments.

Author reply: Thank you for your comment. We add information about the culture conditions in the revised manuscript.

  1. There is no information in the materials and methods regarding cell density in the suspension administered to experimental animals.

Author replyThank you for your comment. We add information about cell density in the suspension administered to experimental animals in the revised manuscript. MDA-MB-231-Luc cells (5×106) were implanted subcutaneously into 24 female nude mice.

  1. Were the same concentration of chemotherapeutic agents used in the in vivo study as in the in vitrostudy? How long were the animals given drugs? Please complete this data in the manuscript.

Author reply: Thank you for your comment. To investigate the inhibitory effect of flubendazole in combination with paclitaxel on BC, we employed cytotoxicity assays in vitro. The results indicated that that the optimum combination ratio of flubendazole and paclitaxel is 3:1 with a concentration of 1.5 μM and 0.5 μM, respectively, based on the analysis of the CI and IC50 values. For the evaluation of in vivo levels, through literature and preliminary experiments, we adopted a concentration of 15mg/Kg flubendazole and 5 mg/Kg paclitaxel.

  1. 6. In "in vitrostudies", I suggest adding the results of examining the effect of the tested combination of chemotherapeutic agents on normal cells, e.g. fibroblasts. The presented experiments are well planned, however, their weakness is the lack of reference to experiments determining cytotoxicity on normal cells. If there are no such results, we do not know whether the combination of drugs used is not more toxic to normal cells of the body. I know that the effects of drugs on normal cells have been evaluated in vivo, but I believe that such in vitrodata should also be included in the manuscript.

Author reply: Thank you for your comment. In the revised manuscript, we have added the cytotoxicity on normal cells (MCF-10A) by each group. The results show that at the doses in the manuscript, flubendazole combined with paclitaxel has no obvious toxic effect on MCF-10A cells.

Reviewer 2 Report

This is an interesting article exploring the synergistic action of paclitaxel and an anthelmintic drug. The experimental design is well-structured. Below you will find some comments:

The manuscript needs language improvements. Please read it carefully and correct mistakes/improve sentences for better understanding.

Introduction

1.      There are minor mistakes in the introduction as well as major language mistakes that make it sometimes difficult for the reader to understand some sentences.

Also, the introduction seems to be” rushed” with no logical connection between the sentences.

Please improve the introduction.

Minor additions

2.      Line 36: report the class of drugs that paclitaxel belongs to and briefly the mechanism of action (2 lines).

3.      Line 40: “CapG” explain what it is.

4.      It would be useful to report the mechanism of action of Flubendazole in the introduction as it can explain why it has an antiproliferative effect.

Results

5.      Correct lines 70-71.

6.      Line 72: “proliferation size”, what do you mean??

7.      In the figures you compare the synergistic effect of the two drugs with the control group. Is there a statistical significance for the comparison of the synergistic effect of both drugs with each drug alone? In order to suggest that flubendazole enhances the inhibitory effect of paclitaxel, the correlation of the synergistic effect should be compared to paclitaxel group.

8.      In Figure 2, the panels for Western (Q, R) are missing.

9.      Line 255: The Fig. 7A-B is wrong. The results in this line are shown in Fig. 7G.

Major comment

10.   Flubendazole is indicated as a major regulator of autophagy. In addition, PI3K/AKT inhibitors can induce autophagy in different cancers. Your results suggest that the enhancing effect of flubendazole is associated with the inhibition of PI3K/AKT pathway. Please check the expression of autophagy markers, LC3-II, SQSTM1/p62, p-ULK1-ser758, ATG5.

Discussion

11.   The discussion needs improvement. Many of the information on the first paragraph could be transferred in the introduction to explain the rationale of this study, and some of them should remain in the discussion to support the major conclusion that flubendazole acts synergistically with paclitaxel (lines 295-30).

12.   There is a very recent paper published which should be discussed in this section.

https://doi.org/10.1016/j.ijpharm.2023.123324

13.   The bibliography for flubendazole should be further discussed. How does your study support or not previous published data? Elaborate more.

14.   Also add and discuss the following: doi: 10.18632/oncotarget.3436

Material and methods

 15.   This section must be improved. The way of presenting the methodology is unusual. Language corrections are also needed.

The manuscript needs language improvements. Please read it carefully and correct mistakes/improve sentences for better understanding.

Author Response

Reviewer 2

This is an interesting article exploring the synergistic action of paclitaxel and an anthelmintic drug. The experimental design is well-structured. Below you will find some comments:

The manuscript needs language improvements. Please read it carefully and correct mistakes/improve sentences for better understanding.

Author reply: Thank you for your good advice. We have carefully revised the language of the manuscript.

Introduction

  1. There are minor mistakes in the introduction as well as major language mistakes that make it sometimes difficult for the reader to understand some sentences.

Also, the introduction seems to be” rushed” with no logical connection between the sentences. Please improve the introduction.

Author reply: Thank you for your comment. We have carefully revised the manuscript to improve the clarity and readability of our paper. We have moved the introduction of PI3K/AKT related mechanisms to the discussion section and added more relevant content to connect the context of the introduction section. For example, we have added “Therefore, it is of great significance to explore new synergistic drug combinations to overcome the side effects and drug resistance of paclitaxel in breast cancer cells” to connect chemotherapy and paclitaxel.

Minor additions

  1. Line 36: report the class of drugs that paclitaxel belongs to and briefly the mechanism of action (2 lines).

Author reply: Thank you for your comment. We added relevant introduction to paclitaxel in the revised manuscript. For example, “Paclitaxel, as a first-line chemotherapy drug for BC, is believed to induce allergy and neuropathy at high doses, which is the main side effect of paclitaxel.”

  1. Line 40: “CapG” explain what it is.

Author reply: Thank you for your comment. Macrophage-capping protein (CapG, also known as gCap39 or MCP) is a member of the gelsolin superfamily which plays important roles in regulating actin assembly. A study revealed that CapG promotes the drug resistance of BC to paclitaxel by medi-ating PI3K/AKT signaling pathway, accompanied by poor prognosis of BC patients [1].

[1] Chi, Y.; Xue, J.; Huang, S.; Xiu, B.; Su, Y.; Wang, W.; Guo, R.; Wang, L.; Li, L.; Shao, Z.; et al. CapG promotes resistance to paclitaxel in breast cancer through transactivation of PIK3R1/P50. Theranostics 2019, 9, 6840-6855.

  1. It would be useful to report the mechanism of action of Flubendazole in the introduction as it can explain why it has an antiproliferative effect.

 Author reply: Thank you for your constructive suggestions. We added a relevant introduction to flubendazole in the revised manuscript. For example, “Our previous studies have indicated that flubendazole is a potential chemical for the treatment of BC through regulating autophagy.”

Results

  1. Correct lines 70-71.

Author reply: Thank you for your comment. We have corrected this error in the revised manuscript.

  1. Line 72: “proliferation size”, what do you mean??

Author reply: Thank you for your comment. We have corrected this error in the revised manuscript. For example, we have replaced “proliferation size” with “proliferation”.

  1. In the figures you compare the synergistic effect of the two drugs with the control group. Is there a statistical significance for the comparison of the synergistic effect of both drugs with each drug alone? In order to suggest that flubendazole enhances the inhibitory effect of paclitaxel, the correlation of the synergistic effect should be compared to paclitaxel group.

Author reply: Thank you for your comment. Statistical analysis used SPSS software version 25.0 and GraphPad Prism 8.0 software. The mean values of the two groups of independent samples were compared by t-test. The difference between each group was analyzed by one-way ANOVA, and then Dunnett's post - hoc test was performed. P<0,05, the difference was considered statistically significant. Moreover, compared with the normal group, the paclitaxel alone group has a certain inhibitory effect on tumor cells (p < 0.05 or p < 0.01), while the p < 0.001 of the combined group.

  1. In Figure 2, the panels for Western (Q, R) are missing.

Author reply: Thank you for your comment. We have corrected this error in the revised manuscript.

  1. Line 255: The Fig. 7A-B is wrong. The results in this line are shown in Fig. 7G.

Author reply: Thank you for your comment. We have corrected this error in the revised manuscript.

Major comment

  1. Flubendazole is indicated as a major regulator of autophagy. In addition, PI3K/AKT inhibitors can induce autophagy in different cancers. Your results suggest that the enhancing effect of flubendazole is associated with the inhibition of PI3K/AKT pathway. Please check the expression of autophagy markers, LC3-II, SQSTM1/p62, p-ULK1-ser758, ATG5.

Author reply: Thank you for your comment. we explored the effect of flubendazole on drug-resistant strains in MCF-7/PTX cells and tested LC3-II, SQSTM1/p62 and ATG5 expression. The results show that flubendazole, as an autophagy inducer, can inhibit the PI3K/AKT signaling pathway.

Discussion

  1. The discussion needs improvement. Many of the information on the first paragraph could be transferred in the introduction to explain the rationale of this study, and some of them should remain in the discussion to support the major conclusion that flubendazole acts synergistically with paclitaxel (lines 295-30).

Author reply: According to your suggestions, we have added relevant discussion “Taken together, these findings highlight the synergistic anti-tumor effects of combining paclitaxel with flubendazole, representing a valuable approach to enhance therapeutic efficacy and overcome resistance to paclitaxel monotherapy in BC.” in the revised manuscript.

  1. There is a very recent paper published which should be discussed in this section.

https://doi.org/10.1016/j.ijpharm.2023.123324

Author reply: Thank you for your comment. We have read and cited the recent paper to enrich this manuscript. For example, “Interestingly, orally inhalable flubendazole nanocrystals also synergistically enhance the cytotoxicity of paclitaxel on A549 lung cancer cells, reduce adverse reactions, and solve the problems of poor solubility and bioavailability of flubendazole in clinical practice.”

  1. The bibliography for flubendazole should be further discussed. How does your study support or not previous published data? Elaborate more.

Author reply: Thank you for your constructive suggestions. Combining previous research findings with our own, our manuscript provides a detailed comparison and discussion outlook in the discussion section. For example, “Consistent with previous findings, this study demonstrates that the combination of flubendazole and paclitaxel can have a synergistic effect on BC cells. In this regard, we demonstrate that the combination therapy has a remarkable inhibitory effect on BC, associated with the significant negative impact on microtubule dynamics.”

  1. Also add and discuss the following: doi: 10.18632/oncotarget.3436

Author reply: Thank you for your comment. We have read and cited this paper to enrich this manuscript. For example, “In addition, some studies have found that flubendazole can enhance the cytotoxicity of fluorouracil and adriamycin by inhibiting tubulin polymerization and inducing the formation of monopolar spindle in breast cancer.”

Material and methods

  1. This section must be improved. The way of presenting the methodology is unusual. Language corrections are also needed.

Author reply: Thank you for your comment. We have carefully revised the manuscript to improve the clarity and readability of our paper. For example, “MCF-7 and MDA-MB-231 originated from ATCC and were maintained in Dul-becco’s Modified Eagle Medium (DMEM) containing 10% fetal bovine serum (FBS) and 1% penicillin–streptomycin (Life Technologies) at 37 °C in 5% CO2.”

Round 2

Reviewer 1 Report

Thank you for the authors' answers. I recommend the revised manuscript for publication in IJMS.

Author Response

Author reply: Thank you for your comment.

Reviewer 2 Report

The manuscript has been improved according to the comments, but there are still issues to be addressed:

a. In general there are still mistakes in the manuscript. Double or missing words and extra spaces. For example, in the introduction, line 29, the word "cells" is missing.

Lines 64-65:

"Accordingly, a concentration of 1.5 μM flubendazole, 0.5 μM paclitaxel and 1.5 μM flubendazole and 0.5 μM paclitaxel were used in this study."

there is a repetition.

b. My comment (no.3) from the first report hasn´t been answered by the authors.

In the figures, you compare the synergistic effect of the two drugs with the control group. Is there a statistical significance for the comparison of the synergistic effect of both drugs with each drug alone? In order to suggest that flubendazole enhances the inhibitory effect of paclitaxel, the correlation of the synergistic effect should be compared to paclitaxel group.

It is not clear what is the control group. And the most significant thing is that if you want to suggest that flubendazole enhances the inhibitory effect of paclitaxel, you have to compare the combination group (Flubendazole + paclitaxel) with the paclitaxel group and show that there is a statistical significant difference between those two group, not only with the control group.

c. The panels Q, R are still missing from figure 2. Please add them.

d. In Figure 3 (E-H) the G2/M phase in the Paclitaxel group and the combination group seem to have almost the same bars for MDA-MB-231 cells and it is slightly prolonged in the combination group for the MCF7 cells. But in the manuscript, you write:

"Flow cytometry analysis showed that after treatment of TNBC cells with
flubendazole and paclitaxel, G2/M phase was significantly prolonged to varying degrees, and the proportion of G2/M phase in the combined application group was significantly increased (Fig. 3E-H), suggesting that the combined application of the two drugs led to tumour cell cycle arrest."

Please elaborate more.

e. In results section 2.5 you have to comment about the autophagy results. Also, this finding has to be discussed in the discussion section.

f. The section of material and methods must be improved.

The manuscript can be improved.

Author Response

The manuscript has been improved according to the comments, but there are still issues to be addressed:

  1. In general there are still mistakes in the manuscript. Double or missing words and extra spaces. For example, in the introduction, line 29, the word "cells" is missing.

Lines 64-65:

"Accordingly, a concentration of 1.5 μM flubendazole, 0.5 μM paclitaxel and 1.5 μM flubendazole and 0.5 μM paclitaxel were used in this study."

there is a repetition.

Author reply: Thank you for your comment. We have replaced “Recurrence of metastatic breast cancer is characterized by rapid proliferation of drug-resistant cancer and high mortality.” with “Recurrence of metastatic breast cancer is characterized by rapid proliferation of drug-resistant cancer cells and high mortality.”, and replaced “Accordingly, a concentration of 1.5 μM flubendazole, 0.5 μM paclitaxel and 1.5 μM flubendazole and 0.5 μM paclitaxel were used in this study” with “Accordingly, a concentration of 1.5 μM flubendazole, 0.5 μM paclitaxel, combined ap-plication group of 1.5 μM flubendazole and 0.5 μM paclitaxel were used in this study”. Moreover,we also checked the full text of the manuscript and corrected any errors.

  1. My comment (no.3) from the first report hasn´t been answered by the authors.

In the figures, you compare the synergistic effect of the two drugs with the control group. Is there a statistical significance for the comparison of the synergistic effect of both drugs with each drug alone? In order to suggest that flubendazole enhances the inhibitory effect of paclitaxel, the correlation of the synergistic effect should be compared to paclitaxel group.

It is not clear what is the control group. And the most significant thing is that if you want to suggest that flubendazole enhances the inhibitory effect of paclitaxel, you have to compare the combination group (Flubendazole + paclitaxel) with the paclitaxel group and show that there is a statistical significant difference between those two group, not only with the control group.

Author reply: Thank you for your comment. In the revised manuscript, we compared the combination group (flubendazole + paclitaxel) with the paclitaxel group, and the results showed that flubendazole enhances the inhibitory effect of paclitaxel.

  1. The panels Q, R are still missing from figure 2. Please add them.

Author reply: Thank you for your comment. In the revised manuscript, we have added the missing Figure. 2Q-R.

  1. In Figure 3 (E-H) the G2/M phase in the Paclitaxel group and the combination group seem to have almost the same bars for MDA-MB-231 cells and it is slightly prolonged in the combination group for the MCF7 cells. But in the manuscript, you write:

"Flow cytometry analysis showed that after treatment of TNBC cells with
flubendazole and paclitaxel, G2/M phase was significantly prolonged to varying degrees, and the proportion of G2/M phase in the combined application group was significantly increased (Fig. 3E-H), suggesting that the combined application of the two drugs led to tumour cell cycle arrest."

Please elaborate more.

Author reply: Thank you for your comment. We have replaced “Flow cytometry analysis showed that after treatment of TNBC cells with flubendazole and paclitaxel, G2/M phase was significantly prolonged to varying degrees, and the proportion of G2/M phase in the combined application group was significantly increased (Fig. 3E-H), suggesting that the combined application of the two drugs led to tumour cell cycle arrest.” with “. Flow cytometry analysis showed that after treatment of TNBC cells with flubendazole and paclitaxel, G2/M phase was significantly prolonged to varying degrees in MDA-MB-231 cells and MCF-7 cells, and the proportion of G2/M phase in the com-bined application group was increased compared to the flubendazole group and paclitaxel group in MCF-7 cells. (Fig. 3E-H), suggesting that the flubendazole and paclitaxel led to tumor cell cycle arrest, and the combined application group could induce stronger tumor cell cycle arrest in MCF-7 cells.”

  1. In results section 2.5 you have to comment about the autophagy results. Also, this finding has to be discussed in the discussion section.

Author reply: Thank you for your constructive suggestions. We have added autophagy related analysis and discussion in section 2.5 and Discussion. Interestingly, flubendazole inhibited the proliferation of various types of cancer cells through regulating autophagy [53-55]. In addition, our previous research found that flubendazole could treat breast cancer by regulating autophagy [13-15]. In this study, we also found that flubendazole can increase the expression level of autophagy related factors and induce autophagy in MCF-7/PTX cells. It is suggested that flubendazole might induce autophagy to treat breast cancer. However, it is necessary to investigate the mechanisms behind future investigations.

  1. The section of material and methods must be improved.

Author reply: Thank you for your comment. We have carefully revised the experimental methods section and added the corresponding references.

Round 3

Reviewer 2 Report

I thank the authors for addressing my comments.The manuscript was significantly improved and I recommend to be published in the current form.

The manuscript improved significantly.